# Impact of Freeze–Thaw Cycles on the Long-Term Performance of Concrete Pavement and Related Improvement Measures: A Review

**DOI:** 10.3390/ma15134568

**Published:** 2022-06-29

**Authors:** San Luo, Tianwen Bai, Mingqin Guo, Yi Wei, Wenbo Ma

**Affiliations:** 1Guizhou Road and Bridge Group Cooperation Ltd., Guiyang 550000, China; sanluogzlq@163.com; 2College of Civil Engineering and Mechanics, Xiangtan University, Xiangtan 411105, China; 202021002624@smail.xtu.edu.cn (T.B.); guomingqin74@xtu.edu.cn (M.G.); 201905800421@smail.xtu.edu.cn (Y.W.)

**Keywords:** freeze–thaw cycles, multi-factor coupling, deicing agents, concrete pavement

## Abstract

Freeze–thaw damage is one of the most severe threats to the long-term performance of concrete pavement in cold regions. Currently, the freeze–thaw deterioration mechanism of concrete pavement has not been fully understood. This study summarizes the significant findings of concrete pavement freeze–thaw durability performance, identifies existing knowledge gaps, and proposes future research needs. The concrete material deterioration mechanism under freeze–thaw cycles is first critically reviewed. Current deterioration theories mainly include the hydrostatic pressure hypothesis, osmolarity, and salt crystallization pressure hypothesis. The critical saturation degree has been proposed to depict the influence of internal saturation on freeze–thaw damage development. Meanwhile, the influence of pore solution salinity on freeze–thaw damage level has not been widely investigated. Additionally, the deterioration mechanism of the typical D-cracking that occurs in concrete pavement has not been fully understood. Following this, we investigate the coupling effect between freeze–thaw and other loading or environmental factors. It is found that external loading can accelerate the development of freeze–thaw damage, and the acceleration becomes more evident under higher stress levels. Further, deicing salts can interact with concrete during freeze–thaw cycles, generating internal pores or leading to crystalline expansion pressure. Specifically, freeze–thaw development can be mitigated under relatively low ion concentration due to increased frozen points. The interactive mechanism between external loading, environmental ions, and freeze–thaw cycles has not been fully understood. Finally, the mitigation protocols to enhance frost resistance of concrete pavement are reviewed. Besides the widely used air-entraining process, the freeze–thaw durability of concrete can also be enhanced by using fiber reinforcement, pozzolanic materials, surface strengthening, Super Absorbent Polymers (SAPs), and Phase Change Materials. This study serves as a solid base of information to understand how to enhance the freeze–thaw durability of concrete pavement.

## 1. Introduction

Currently, the infrastructure construction in China is under heavy development. Pavement construction is essential for building a highly efficient transportation system. Concrete is currently the most widely used manufactured material, and its wide application has reshaped modern urban construction. Concrete has also been widely applied in pavement construction, and the Continuously Reinforced Concrete Pavement (CRCP) has the longest service life among all types of pavement structures [1]. The total road mileage in China has reached more than 5.2 million kilometers in 2022, and concrete pavement accounts for around 60% of the mileage [2]. It should be noted that the majority of concrete pavement was built before the 21st century. According to relevant information and surveys by construction departments [3,4,5,6], these pavement constructions may not fulfill the current durability and service life requirements. The durability of concrete pavement is essential for transportation system efficiency. The durability of concrete structures is the ability of concrete to resist environmental loading and maintain its good serviceability and integrity over time. China has entered a period of frequent maintenance for concrete pavement in the 21st century [7]. It is foreseeable that durability design will become increasingly crucial in concrete pavement design in the future.

Compared to asphalt pavement, concrete pavement has the advantages of resistance to water damage due to its hydraulic nature [8] and superior rutting resistance due to its strong stiffness [9]. However, concrete pavement can still deteriorate under service in harsh environments, including freezing and in coastal regions. Freeze–thaw can lead to severe degradation of the mechanical performance of concrete pavement and reduce its service life. Freeze–thaw damage is one of the most concerning durability issues for concrete pavement in North China, Northwest China, and Northeast China. Extensive freeze–thaw issues occurs in almost 100% of concrete pavement in Northeast China [10]. In addition to the three northern regions mentioned above, freeze–thaw damage can also be observed in areas north of the Yangtze River in eastern and central China and in the cold regions of southwest China [10]. Freeze–thaw damage can easily lead to surface spalling of concrete pavement. The developed cracks facilitate the penetration of water and harmful ions, leading to corrosion of reinforcement properties. The loading capacity, fatigue cracking-resistance, and service life can be significantly reduced due to the developed freeze–thaw damage and coupled durability issues. Freeze–thaw cycles can lead to the unique “D-cracking” that occurs in concrete pavement, as shown in Figure 1, which is caused by the cracking of aggregate that is susceptible to freeze–thaw damage. Hence, it is essential to investigate the concrete pavement’s freeze–thaw durability thoroughly.

Since its invention in 1824, Portland cement concrete has been used for almost 300 years. In addition to the mechanical performance, the durability performance of Portland cement concrete is essential in the long-term performance of concrete infrastructure. However, the freeze–thaw damage mechanism for concrete has not been fully understood. Currently, the most recognized freeze–thaw damage theories are the hydrostatic pressure hypothesis proposed by Powers in 1945 [12] and 1949 [13], and the osmotic pressure hypothesis proposed by Helmuth in 1953 [14]. The ice prism theory, hydrostatic pressure correction theory (based on supercooled liquids), and the saturation theory have also been proposed [15,16,17]. In the 1990s, Scherer analyzed the coexistence mechanism of gas–liquid–solid three-phase equilibrium from the perspective of thermodynamics [18,19] and proposed the theory of ice expansion pressure. However, most of these classical models were obtained from purely physical assumptions, and some were only obtained from tests with net cement or mortar specimens [20]. The level of durability of concrete in practical engineering is generally caused by coupled factors. Concrete pavement is often subjected to the coupling effects of freeze–thaw damage, harmful ion penetration, and corrosion of reinforcement, carbonation, and fatigue loading. A coupling effect can further accelerate the development of freeze–thaw damage. For example, deicing salts widely used on concrete pavement for snow removal during winter. Deicing salt can lead to severe freeze–thaw damage due to increased crystal expansion pressure and can accelerate reinforcement corrosion [21,22].

Currently, 19 review papers can be found on the “webofscience” database related to the freeze–thaw damage of concrete pavement. These papers mainly focus on the deicing agent in concrete, the coupling damage by both freeze–thaw and external loading [23], the freeze–thaw concrete damage containing phase-change materials [24,25] and recycled materials [26,27,28,29,30,31,32], or deicing salts [33]. Several papers related to the freeze–thaw of pavement focuses on the durability performance of previous concrete [34] or roller-compacted concrete pavement [35]. Overall, a critical review on the freeze–thaw damage of concrete pavement is still quite limited.

Focusing on the issues of freeze–thaw durability of concrete pavement, this study seeks to summarize the current understanding of the freeze–thaw mechanism of concrete materials. Recent investigations on the freeze–thaw damage of concrete pavement are summarized in Table 1. The coupling correlation between freeze–thaw damage and related correlated durability issues is analyzed. Finally, the influence of freeze–thaw damage on the long-term mechanical performance of concrete pavement is investigated. The corresponding damage mitigation protocols are proposed. The scheme of the entire article structure is shown in Figure 2.

## 2. Theory of Mechanical Deterioration of Concrete under Freeze–Thaw Action and Associated Damage

Researchers started to identify the existence of freeze–thaw damage in the 1930s, and numerous investigations have been conducted since. This section mainly aims to summarize the current understanding of freeze–thaw damage and to identify the existence of knowledge gaps on freeze–thaw damage to concrete pavement. As summarized in reference [51], the mass loss [52,53] of concrete due to surface spalling can be generated during freeze–thaw cycles and the developed internal cracking can lead to strength deterioration [54]. Further, the specific heat capacity of concrete can be decreased [55], while the thermal conductivity [56] and the coefficient of thermal expansion [57,58] can be increased under freeze–thaw cycles. Due to the initiation and propagation of internal deterioration under freeze–thaw cycles, the porosity of the concrete will increase due to a coarsening effect [59]. Freeze–thaw damage theories have been proposed to analyze the mechanisms of the mechanical and thermophysical properties of concrete.

### 2.1. Hydrostatic Pressure Hypothesis

In 1946, Powers proposed the classical hydrostatic pressure hypothesis based on observed phenomena occurring in concrete during freezing and swelling. The theory suggests [12] that when the temperature drops to a certain level, a portion of the pore water in the concrete pore space will change from liquid water to solid ice and expand by 9% in volume. This will force the still unfrozen pore solution in the frozen zone to migrate towards the unfrozen zone. During this process, hydrostatic pressure is generated and acts in the surrounding matrix. The hydrostatic pressure increases with the length of the pore water flow, so there is a limited flow length in the tensile strength of the concrete. When the pore water flow is more significant than this limited flow length, the resulted hydrostatic pressure exceeds the tensile strength of the cement matrix, resulting in the initiation and prorogation of micro-cracks. Under continuous freeze–thaw cycles, the cracks caused by the damage will slowly accumulate, and the pores will become connected. The accumulated micro-scale damage will finally lead to macro-scale deterioration of the concrete materials. Based on Powers’ theory, the freeze–thaw damage cannot occur when pore saturation is lower than 91.7%. However, freeze–thaw damage can be observed in concrete with a saturation rate of 70–80% [60]. Besides that, it is difficult to explain the shrinkage of air-entrained concrete and the continuous swelling of non-air-entrained concrete samples under freeze–thaw cycles [61].

Based on Powers’ hydrostatic pressure hypothesis, Fagerlund [62] further proposed a hydrostatic pressure model as shown in Figure 3. The built theory model for calculating the maximum hydrostatic pressure is indicated in Equation (1). The maximum hydrostatic pressure generated by icing is inversely proportional to the permeability coefficient of the concrete matrix. It is directly proportional to the square of the air bubble spacing, the cooling rate, and the icing rate. Therefore, it is considered that incorporating air-entraining agents into concrete can generate closed air bubbles and decrease the maximum expansion pressure by lowering the permeability coefficient and air bubble distance. Thus, the frost-resistance of concrete materials can be enhanced through air-entrainment.

The d is the distance between the two air bubbles. Assuming the distance from the left air bubble to the freezing point A is x, then the maximum hydrostatic pressure occurs x = *d*/2 at the place. Its calculation formula is:(1)$$p_{\max}=\frac{0.09}{8k}\cdot \frac{dw_{f}}{d\theta }\cdot \frac{d\theta }{dt}\cdot d^{2}$$
where the *k* is the coefficient of the permeability of the material, the *dw_f_/dθ* is the freezing rate, i.e., the increment of frozen water for every 1 °C decrease in temperature. The *dθ/dt* is the cooling rate.

### 2.2. The Osmolarity Hypothesis

Although the hydrostatic pressure hypothesis can explain many phenomena in the freeze–thaw process of concrete (such as the effect of the cooling and freezing rate on the frost damage of concrete or the mechanism of action of air-entraining agents, etc.), several critical experimental phenomena cannot be well explained, including the fact that organic liquids which do not expand in volume after freezing can also cause damage to concrete. Therefore, Powers and Helmuth et al. further developed the osmotic pressure hypothesis. The osmotic pressure hypothesis indicates [14,63] that the concrete pore solution is weakly alkaline and contains multiple-element ions. As a result, the salt in a concrete pore solution can lead to the formation of ice crystals at hostile temperatures and can cause an increase in the concentration of the unfrozen solution in the frozen zone. The concentration difference between the concrete pore solution in the frozen area and the solution in the unfrozen zone can then be generated, resulting in osmotic pressure. When the osmotic pressure increases to a certain level, the tensile force formed will exceed the tensile strength of the substrate, causing cracks within the substrate and creating frost damage.

### 2.3. Theory of Critical Saturation

It has been widely confirmed that freeze–thaw damage development depends on the saturation rate of the concrete. Fagerlund first proposed the critical satiation theory in 1977, which describes a substantial satiation value for porous materials. Once this threshold is exceeded, the material can only withstand one freeze–thaw cycle before cracking or becoming damaged [64,65]. This theory is based on the hydrostatic pressure theory, and Chen’s research [66] suggests that critical satiation is a physical property of a material that does not depend on environmental factors. Suppose the external environmental effects of different concretes are converted to satiation. In this case, the difference between the satiation and the concrete’s critical satiation could be used to measure the concrete’s resistance to freezing in a particular environment.

### 2.4. The Salt Crystallization Pressure Hypothesis

The crystallization pressure originates from the growth of ice from large pores to small pores. A hemispherical interface is formed between ice and water due to a capillary effect. The pressure is generated due to the difference in the ice–water interface curvature and the cylindrical pore curvature. The crystallization pressure is then needed to balance the pressure caused by the curvature difference [67]. According to an investigation by Scherer [18], the maximum driving force for crystallization can be divided into salt crystals grown that are driven by supersaturation, and ice crystals increased that are driven by supercooling. The pressure generated on the walls of concrete pores subjected to frost damage can be affected by the pore size and the interfacial energy between the pore wall and the crystals. It is generally known that the freezing point of the pore solution decreases with the pore size [68]. The water movement in the pore system is restricted by ice generated in the large pores and the resulting hydraulic pressure. The crystallization pressure exceeds the concrete tensile strength under a subcooling state [69,70], and cracks are then generated.

### 2.5. Summary on the Freeze–Thaw Damage Mechanism

In addition to the four hypotheses introduced above, the pore structure, temperature stress, ice prism, micro-ice-lens [71], salt freeze [72], and hydrostatic pressure correction based on supercooled liquids, are essential parts of the theoretical system of concrete freeze–thaw damage mechanisms. Currently, the investigation of the deterioration mechanism of concrete under freeze–thaw cycles mainly focuses on the deterioration initiated by the cement matrix. However, the typical D-cracking in concrete pavement is caused by the cracking of aggregate susceptible to freeze–thaw damage [73,74]. D-cracking is caused by the moisture absorption of certain coarse aggregate, including the Kansas limestone aggregate [75]. D-cracking is considered to be a progressive structural deterioration in concrete pavement. 

Currently, the investigation of the cracking mechanism of these aggregates is not sufficient. For example, the deicing salt used for snow removal on the surface of concrete pavement can increase the concrete pore solution’s salinity, and thus change its freezing temperature. The expansion pressure during freezing and the deterioration mechanism of concrete material may then also be altered. Further investigation is needed to understand the deterioration mechanism of concrete material in the pavement.

## 3. Coupling of Freeze–Thaw Damage and Other Factors

Concrete pavement in actual service conditions is usually subjected to various complex environmental factors. Hence, concrete pavement can deteriorate under a coupled action effect and co-create durability issues [76,77].

### 3.1. Coupling of Freeze–Thaw Cycles and External Loads

Concrete pavement bears continuous transportation loading during its service life. Current research shows that the coupling effect of external loading and freeze–thaw cycles can lead to more severe deterioration of the relative dynamic modulus concrete materials and accelerate the deterioration process [78]. The microscopic mechanism of the coupling effect of freeze–thaw cycles and loading on the durability performance of concrete includes two results [77]. The microstructural damage caused by external loading can accelerate freeze–thaw wear. Meanwhile, the non-uniform deterioration generated by freeze–thaw leads to stress redistribution and microstructural path changes during the loading process. The coupling effect can also be affected by the level of external loading. Under relatively large loading, the microcracks caused by freeze–thaw can easily lead to stress concentration, and the propagation of microcracks can rapidly damage the concrete. Under relatively small loading, concrete materials are mainly deteriorated by the formed uniform microcracks, and the deterioration rate is relatively low. Furthermore, the freeze–thaw resistance of concrete can be lowered with higher stress levels [79]. At the same time, it has also been shown that the coupling effect can be affected by the loading type [80]. Former studies indicate that sudden brittle damage can be observed in the freeze–thaw-damaged concrete under bending loads. Currently, the information about the influence of the loading type on the coupling effect is not sufficient, and further research is needed.

### 3.2. Coupling of Freeze–Thaw Cycles with Different Salt Solution Environments

According to the freeze-swelling mechanism and related experiments, it is found that the mass-loss rate and decreasing rate of dynamic modulus can be increased by freeze–thaw cycles of concrete in a salt solution [81]. At the beginning of freezing, the salt solution lowers the freezing point of the concrete pore solution, thus slightly slowing down the decrease in the relative dynamic elastic modulus of concrete than under freeze–thaw action alone. For example, in the early stages of the test, a sulphate attack can have an inhibitory effect on the freeze–thaw damage of hydraulic concrete, resulting in a certain degree of improvement in the durability indexes. However, in the latter stages of the test, the declination rate in the durability indexes of hydraulic concrete increases significantly under the coupling effect of a sulphate attack and freeze–thaw cycles [82]. The mechanism of salt solution coupling on surface scaling mainly includes two aspects. Firstly, the water absorption rate (degree of water saturation) of concrete in a salt solution increases compared to pure water, thus intensifying the freeze–thaw damage. Additionally, the salt solution in the pores of the concrete becomes supersaturated. It crystallizes after the water evaporates and dries, producing a higher crystallization pressure which also causes the concrete to swell and crack.

At the same time, experiments have shown that during the freeze–thaw cycle [83], different salt solutions have different degrees of influence on the relative dynamic elastic modulus damage rate and the deterioration process of high-strength concrete, detailed in the following order: (5% MgCl_2_ + 5% Na_2_SO_4_) complex solution < 5% MgSO_4_ solution < 10% NaCl solution. The salt for snow removal on concrete pavement mainly includes NaCl, KCl, CH₃COONa, and CH₃COOK. The investigation of the coupling effect between freeze–thaw and attack by acetates is still limited. The former studies indicate that acetates can lead to an alkali–silica reaction in concrete. Further research is needed to unveil the correlation between acetate exposure and freeze–thaw cycles.

### 3.3. Coupled Effects of Freeze–Thaw Cycles, External Loads, and Salt Solution Erosion

In the cold winter temperature of the marine environment or in cold areas where deicing agents are used, concrete pavement often suffers from the standard coupling of freeze–thaw cycles, external loads, and salt solution erosion. At present, relatively little research has been carried out on the deterioration laws of concrete under the combined effect of these three factors.

Studies have shown [84,85,86] that the form of damage is dominated by surface spalling under tensile and flexural loading, freeze–thaw cycles, and chloride salt attack for ordinary concrete. For high-strength concrete, surface spalling is minimal. The tensile load accelerates the surface spalling and internal damage of concrete during the freeze–thaw cycling–chlorine attack. It significantly affects the relative dynamic modulus of elasticity, particularly causing brittle fracture. Tensile loading hinders chloride ion intrusion into the concrete in the compression zone. Still, the depth of chloride ion intrusion increases significantly in the tension zone, leading to an increased risk of corrosion of the reinforcement in this area. Overall, the bending load accelerates the freeze–thaw cycle–chloride salt attack damage to the concrete and even causes brittle fracture of the concrete.

## 4. Damage to Pavements Subjected to Freeze–Thaw Cycles

Concrete pavement has a superior load-bearing capacity and durability, enabling it to be used worldwide. As mentioned earlier, concrete pavement load-bearing power and durability can be significantly reduced when subjected to a combination of complex factors in the environment. Hence the investigation of concrete pavement is necessary.

### 4.1. Damage Mechanism of Concrete Pavement under the Action of Freeze–Thaw Cycles

At the meso-level, concrete is a three-phase composite material, often considered a mixture of aggregate, cement mortar, and pore combinations. The evolution and spatial distribution of internal random pore and crack structures under the action of freeze–thaw cycles is a critical factor in the study of damage mechanisms and macroscopic mechanical properties of concrete materials [87]. The prevalent damage modes for concrete pavement are surface scaling and internal cracking. Surface scaling mainly occurs at the concrete surface and affects the surface serviceability. The internal cracking causes damage to the strength and modulus of elasticity of the concrete, eventually leading to the deterioration of the entire concrete structure [88,89].

For pavement and bridge decks, freeze–thaw damage often acts in coupling with fatigue loading. For this reason, a fine-scale method based on the rigid body spring method (RBSM) has been conducted [90], which divides the concrete material into mortar, coarse aggregate, and the interface transition zone (ITZ) between them. It has been found that the ITZ quality is essential for the durability performance of concrete pavement [91]. The study results show that the structure’s static strength and fatigue life decrease as the degree of frost damage (irreversible plastic deformation) increases. Besides the damage initiated from the cement matrix, the concrete pavement can also be deteriorated by cracking aggregate susceptible to freeze–thaw damage. The mechanism analysis for the typical D-cracking is currently limited.

### 4.2. Effect of Deicing Agents on Freeze–Thaw Damage in Concrete Pavement

To protect the safety of vehicles traveling on the road and to guarantee the normal transportation efficiency of the road during winter, deicing agents are often used to treat the road surface for snow removal. However, deicing agents often aggravate the freeze–thaw damage of concrete pavement. The cracks and scaling caused by freeze–thaw damage can also lead to the penetration of chloride ions, which can cause rusting of the reinforcement properties [92,93,94,95,96,97], and ultimately lead to a significant shortening of the pavement’s service life. Some studies have shown that spraying deicing agents on concrete pavement causes damage that is approximately ten times more severe and faster than conventional freeze–thaw cycles [98]. The deterioration mechanism of concrete pavement subjected to the coupling action of deicing agents with freeze–thaw processes can be divided into two main parts, namely, chemical and physical mechanisms.

#### 4.2.1. Chemical Mechanisms

On the chemical level, many deicing agents contain chlorine salts as the main component, which can react with concrete materials and cause the products to dissolve and create pores, or generate new salt crystals and increase crystallization pressure within the concrete. NaCl and CaCl_2_ are commonly used in deicing agents and will be used as examples below [99,100]. When the deicing agents contain NaCl, the NaCl causes the dissolution of the Ca(OH)_2_ in the concrete. Therefore, the location where the Ca(OH)_2_ content initially exists becomes a pore within the concrete. The interaction between the salts and the Ca(OH)_2_ content increases the porosity and permeability of the concrete pavement. It leads to an increase in the amount of ice formation during the freeze–thaw process. The specific chemical reaction equation is shown below.
$$2\mathrm{NaCl}+\mathrm{Ca}{\left(\mathrm{OH}\right)}_{2}\to \mathrm{CaC}l_{2}+2\mathrm{NaOH}$$

If the deicing agent contains CaCl_2_, it will react with tricalcium aluminate (C_3_A) to form calcium chloroaluminate (Friedel’s salt). This reaction will lead to the scaling of the concrete over time. The specific chemical reaction equation is shown below:$$\mathrm{CaC}l_{2}+C_{3}A\to C_{3}A\cdot \mathrm{CaC}l_{2}\cdot H_{2}O$$

#### 4.2.2. Physical Mechanisms

A physical process primarily causes the scaling of concrete surfaces caused by deicing agents. Due to the complexity of the concrete freeze–thaw damage mechanism, the actual tool has not yet been fully unveiled. The proposed hypotheses have mainly guided the research on concrete durability [101]. Current available research can be generally categorized into microstructural and macrostructural factors. Microstructural factors are proposed based on the various freeze–thaw deterioration theories described earlier. These include increases in osmotic and water pressure, water movement due to differences in pore solution concentration within the concrete, vapor pressure deficiencies, and the growth of salt crystals. In addition, deicing agents typically increase the saturation of concrete, leading to a higher saturation level internally, thereby increasing the risk of frost damage [102]. At the macro level, some studies have concluded that an inconsistency in temperature and deformation between ice and concrete during the cooling process generates local stresses that can cause scaling of the concrete surface and internal damage. The effect of ice thickness on the scaling of concrete pavement in the underlying layers was also obtained experimentally [103]. The results show that the damage to the material surface generally increases with the ice thickness under the same conditions of freeze–thaw cycles, freezing fluid salt content, and material properties. As shown in Figure 4, the cumulative scaling after three cycles was 74 g (1.3 kg/m^2^) with an initial ice thickness of 1 mm and reached 103 g (1.8 kg/m^2^) with an ice thickness of 10 mm.

The thermal shock hypothesis explains this phenomenon from another macroscopic perspective. It suggests that the dispersed deicing agents cause a sudden drop in the temperature of the concrete surface, which causes a difference in internal and external stresses. The change in the concrete volume caused by the temperature stress results in tensile stresses inside the concrete. If the induced tensile stress is higher than the tensile capacity of the concrete, the concrete will crack [104]. As illustrated in Figure 5, using sodium chloride (NaCl) and calcium chloride (CaCl_2_) as examples, the surface temperature of concrete drops rapidly due to the dissolution of sodium chloride in water and the heat absorption. However, calcium chloride produces an exothermic reaction, which rapidly increases the temperature of the concrete surface. The stresses generated by the rapid change in surface temperature may exceed the tensile strength of the concrete, leading to the development of minor cracks [99].

### 4.3. Coupled Effects of External Loading and Freeze–Thaw Cycles

Concrete pavement needs to resist the impact of wheel loading during its service life; fatigue resistance is essential for long term-durability performance. It is expected the generated internal defects during freeze–thaw cycles can promote crack initiation and propagation under fatigue loading. The development of fatigue cracks can increase water entrance due to enhanced percolation and can accelerate freeze–thaw deterioration. A study by Yang et al. [38] found the coupling effect of fatigue loading and freeze–thaw cycles can significantly reduce the micro-hardness of Interfacial Transition Zone(ITZ) and enlarge the ITZ width, while the influence of fatigue loading alone is not obvious. The development of fatigue loading damage can also be accelerated due to developed freeze–thaw defects which may cause reduced impermeability.

## 5. Damage Mitigation Methods for Concrete Pavements Subjected to Freeze–Thaw Cycles

Freeze–thaw cycles are extremely damaging to concrete pavements, and the annual pavement rehabilitation work is also a considerable burden on national finances [8,105]. Researchers have focused on developing damage-mitigation protocols to enhance the freeze–thaw resistance of concrete pavement. The following are some of the more commonly used as well as some of the new and promising measures that have been developed in recent years. It has been widely confirmed that concrete mix design is essential for freeze–thaw resistance, including the cement content and water–cement ratio [40]. Other new methods have also been proposed for the long-term performance of concrete pavement in frozen areas.

### 5.1. Surface-Strengthening Materials

Surface treatment of concrete to improve its frost resistance is relatively efficient and inexpensive [106]. One study tested the performance of several materials such as concrete protectors, polyurea, epoxy resin, and silane (the main component) in pavement concrete specimens [107]. The results showed that the mass loss and the saturation of the concrete both decreased, and the relative dynamic modulus of elasticity increased after the surface treatment. Among the four concrete surface reinforcement materials, silane significantly improved the water and frost resistance of concrete, enabling it to be a more suitable concrete road surface reinforcement material. A study by Guo and Weng [41] compared the freeze–thaw resistance of concrete pavement treated with silane, modified polyurea, and epoxy resin. The characterization of the physical and mechanical properties of concrete indicate that the reinforcement efficiency from most efficient to least efficient follow the order of silane, modified polyurea, then epoxy resin. The reaction mechanism of the silane protective layer is shown in Figure 6. The network structure formed by the silane condensation reaction is significantly hydrophobic. Its attachment to the surface layer, pores, and micro-cracks of concrete can substantially improve concrete’s water and frost resistance. Although surface-strengthening materials reduce frost damage to concrete pavement, the pavement is still subjected to wear and tear from dynamic loads during its service life. The surface-strengthening material wear and skid resistance need to be tested before it is put into practical use on pavement.

### 5.2. Air-Entraining Agents

Chemical air-entraining agents (AEAs) have long been used as a common and popular measure to mitigate damage to concrete from freeze–thaw cycles. Air-entraining agents stabilize the air bubbles generated during concrete mixing and prevent tiny bubbles from collecting and escaping [108]. These small, well-dispersed micro-bubbles allow pore water to flow during freeze–thaw, thereby reducing the internal stresses in the concrete due to water pressure. Figure 7 briefly depicts the mechanism of action of air bubbles introduced by the air-entraining agent. The initial assumptions are shown in Figure 7a, where the system is idealized as water-free before the concrete specimen incorporated with the air-entraining agent is immersed in water. The light grey area represents the solid space, and the pores—consisting of capillary pores and tiny air bubbles—are represented by the dark grey area. After immersing the sample in water, as shown in Figure 7b, the capillary tube will quickly be filled with water due to capillary action. The saturated pores are represented in blue. The dark grey pore section in Figure 7c indicates that when the concrete suffers freeze–thaw damage, its pore volume can ‘absorb’ the flowing water ‘squeezed’ by the ice, acting as a release zone for the water pressure [109]. The entrained-air bubble network can provide the additional expansion space for the frozen pore solution [110,111]. The spacing factor of the bubble network is also essential for freeze–thaw damage mitigation efficiency [112]. However, air content in concrete containing chemical air-entraining agents (AEAs) is highly variable and influenced by temperature, transport time, mixing, pumping, and internal vibrations [108]. Recently, however, it has been found [108,109,113,114] that super absorbent polymers (SAPs) can effectively improve the freeze–thaw resistance of concrete, acting similarly to air-entraining agents. Many uniformly distributed, closed, and stable micro-pore structures can be generated by the incorporated SAPs into the matrix when releasing water.

### 5.3. Fiber Reinforcement and Modification with Pozzolanic Materials

It has been widely considered fiber reinforcement can enhance the toughness of fragile concrete. It is expected that the introduced fiber can prevent the development of internal cracking and mitigate freeze–thaw damage. A study by Cui et al. [37] indicates that added fiber can effectively enhance the anti-wheel performance of concrete airport pavement under freeze–thaw cycles. Furthermore, the reinforcement efficiency of organic fiber is better than that of inorganic fiber. A study by Haghnejad and Modarres examined the reinforcement effect of polypropylene(PP) fiber on roller-compacted concrete pavement [39]. It was found that reinforcement with PP fiber can obviously enhance the freeze–thaw resistance of roller-compacted concrete pavement based on the examined modulus or rupture. A fiber content of 3% is recommended for engineering practices. A study by Siamardi and Shabani [42] indicates that the fiber content for enhancing freeze–thaw resistance can be lowered to 0.15 by using a micro-scale synthetic fiber. It is found the freeze–thaw resistance of concrete pavement can also be improved by natural jute fiber [43]. PP-fiber-reinforced concrete has been applied in two-lift concrete pavement [46].

In addition to fiber reinforcement, the addition of nanomaterials can densify the internal structure of concrete and improve freeze–thaw resistance by decreasing permeability. A study by Gonzalez et al. [40] enhanced the freeze–thaw resistance of concrete by adding nanosilica with a size of approximately 100 nm. Cement kiln dust can also be used to enhance the freeze–thaw durability of concrete pavement and an optimal amount of 2% is recommended based on durability tests [45]. 

Adding metakaolin and zeolite can effectively reduce the mass loss of concrete caused by surface scaling during freeze–thaw cycles [47]. A study by Zhang et al. [91] indicated that adding silica fume can enhance the quality of ITZ and improve the freeze–thaw resistance of concrete pavement.

### 5.4. Incorporation of Phase Change Materials

Phase change materials (PCM) are latent heat substances that can repeatedly store and release large amounts of thermal energy as they melt and solidify [115,116,117]. Their stored energy can be removed when the ambient temperature drops, thereby delaying or preventing icing [118,119]. In addition to this, the application of phase change materials can help to reduce the use of deicing agents, which can prohibit the damage caused by the coupling effect of deicing salts and freeze–thaw cycles [21,22]. As this protocol does not require additional energy sources [120], the use of phase change materials is more in line with the goal of sustainability than other methods [121,122].

Phase change materials can be divided into three main categories: organic, inorganic, and eutectic materials. Most organic phase change materials do not rely on subcooling for nucleation during freezing and are usually non-corrosive. Inorganic phase change materials can be divided into salt hydrates and metallic phase change materials, each which have their disadvantages. The metallic phase change materials are not in the temperature range required for construction applications, while the salt phase change materials may react with cementitious materials and be highly hazardous to concrete, or worse, may corrode metals [119,120].Both inorganic phase change materials are, therefore, not included in the scope of consideration. Eutectics of organic and inorganic compounds are composed of molten materials of two or more compositions. Therefore, specific mixtures can be designed to match the maximum extent possible of the ambient temperature of the concrete pavement [120].

Figure 8 shows a schematic diagram of phase change materials to melt snow and ice on concrete pavement. The use of phase change materials in concrete infrastructure requires physical, chemical, and thermodynamic properties that are compatible with concrete [123]. The thermodynamic properties include phase change temperatures slightly above 0 °C, high latent heat of fusion, high specific heat (heat capacity), and high thermal conductivity [120,124]. The physical properties of phase change materials should be highly dense, have small volume changes between phases, and maintain low vapor pressure. The chemical properties should be stable, compatible with concrete, nontoxic, and non-flammable [123]. It should also be noted that adding phase change materials in concrete can reduce the mechanical strength due to a relative weak ITZ performance [111]. Hence, the added phase materials should be optimized, while surface treatment is needed to enhance the ITZ performance [121]. Figure 9 shows a schematic diagram of three methods of incorporating phase change materials into concrete. Figure 9a shows a pipe containing phase change material being placed into concrete [123]; Figure 9b depicts PCM capsule particles placed directly into concrete pavement after external encapsulation [125], and Figure 9c shows PCM being filled into concrete pores on a road surface [126].

## 6. Knowledge Gaps and Future Recommendations

Based on a critical review, the identified knowledge gaps are shown below:(1)Currently, the freeze–thaw damage law of concrete pavement has not been fully understood, including the development of the mechanism of D-cracking caused by the specific coarse aggregate, and the coupling effect between freeze–thaw damage and fatigue loading and/or deicing salts. In addition, most investigation is conducted using Portland cement concrete; the deterioration mechanism of concrete pavement prepared with other cementitious materials is quite limited [44].(2)It is still difficult to reveal the differences between indoor test results and field durability performances [98,99,127] due to the difference in material and loading levels. It has been reported that the capillary suction of the lab-prepared concrete is much lower than that of field pavement concrete [128]. A former study also indicates the freeze–thaw damage of concrete pavement can be most severe in the wet-freeze environment [48]. The existing evidence on the influence of environmental relative humidity on the freeze–thaw resistance is not sufficient.(3)Besides the widely used air-entraining methods, the application of recently proposed freeze–thaw damage mitigation protocols have not been widely used in field application. For example, it has been found that the added phase change materials in concrete can reduce its mechanical strength due to relative weak ITZ performance.

Based on the summarized research knowledge gaps, the further research needs are shown below:(1)The development of the mechanism of D-cracking should be investigated by unveiling the correlation between aggregate composition and D-cracking sensitivity. Furthermore, the freeze–thaw deterioration mechanism of low carbon concrete (alkali-activated concrete) should be investigated.(2)The testing methods related to the material and loading-level differences between field exposure and lab tests should be considered. At the material level, the physical (permeability and suction) and mechanical properties of the lab specimen should be consistent with that of the field concrete. At the loading level, the coupling effect of the external loading (fatigue loading) and environmental factors (deicing salts) should be considered in the test design. Furthermore, the correlation between the physical and mechanical performance of concrete and the durability of pavement under coupled fatigue loading and freeze–thaw cycles should be investigated.(3)The field application of the recently proposed freeze–thaw damage mitigation protocols should be conducted, including surface strengthening, the incorporation of phase change materials, and changing absorbency levels of the concrete. Furthermore, the content of the added phased materials and the absorbency should be optimized, while surface treatment is needed to enhance the ITZ performance.

## 7. Conclusions

This study aims first to summarize the current understanding of the freeze–thaw mechanism of concrete materials. The coupling correlation between freeze–thaw damage and related durability issues is analyzed. Finally, the influence of freeze–thaw damage on the long-term mechanical performance of concrete pavement is investigated. The related damage mitigation protocols are proposed. The major research findings are shown below:(1)The current deterioration theories are based mainly on the hydrostatic pressure hypothesis, the osmolarity hypothesis, and the salt crystallization pressure hypothesis. The critical saturation degree has also been proposed to depict the influence of the internal saturation on freeze–thaw damage development. Meanwhile, the influence of pore solution salinity on the freeze–thaw damage level has not been widely investigated. In addition, the deterioration of the typical D-cracking in concrete pavement has not been fully understood.(2)External loading can accelerate the freeze–thaw damage development, and the acceleration can be more obvious under higher stress levels. Further, the salt ions can also interact with the concrete during the freeze–thaw cycles, generating internal pores or leading to crystalline expansion pressure. Freeze–thaw development can be slowed down under relatively low ion concentration due to increased frozen points. The interaction between external loading, environmental ions, and freeze–thaw cycles has not been fully understood.(3)Besides the widely used air-entraining process, the freeze–thaw durability of concrete can be enhanced through Fiber Reinforcement, Surface-Strengthening, and incorporation of Pozzolanic Materials, Super Absorbent Polymers (SAPs), and Phase Change Materials.

## Figures and Tables

**Figure 1 materials-15-04568-f001:**
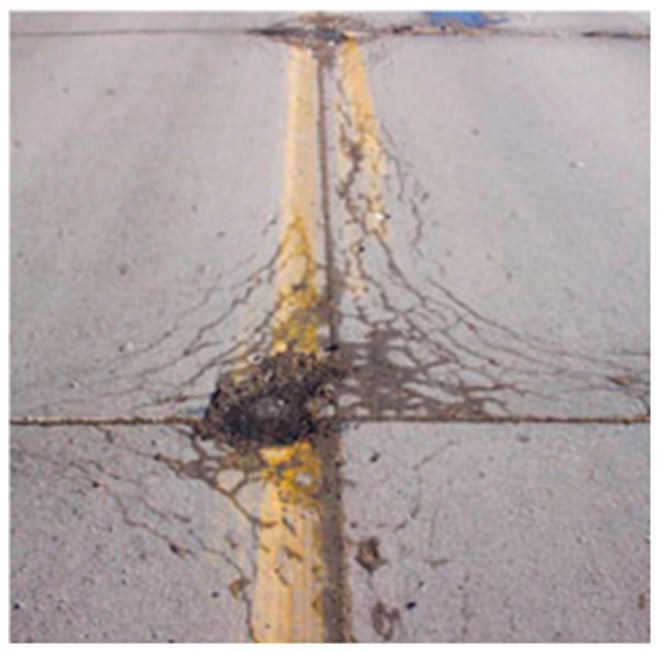
Demonstration of D-cracking in concrete pavement [11].

**Figure 2 materials-15-04568-f002:**
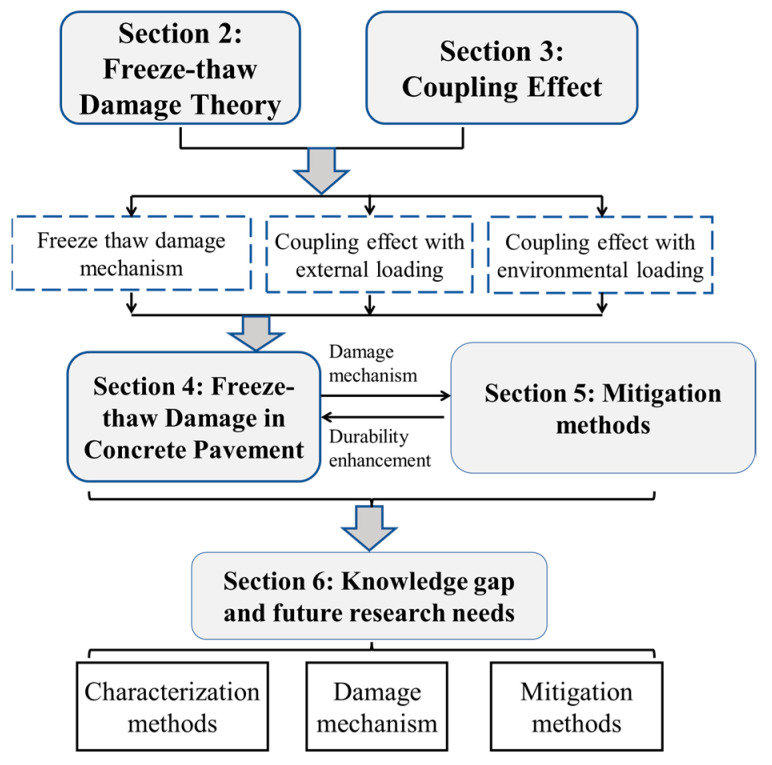
Scheme of the review article.

**Figure 3 materials-15-04568-f003:**
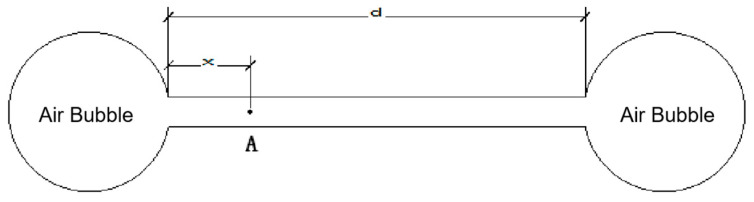
Hydrostatic pressure model.

**Figure 4 materials-15-04568-f004:**
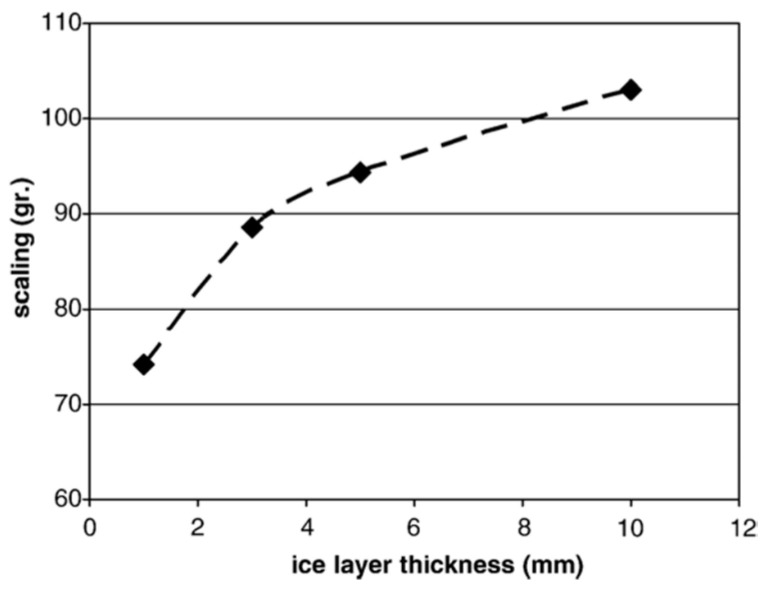
Cumulative mass scaling of identical samples with various ice-layer thicknesses (after three freezing–thawing cycles) [104].

**Figure 5 materials-15-04568-f005:**
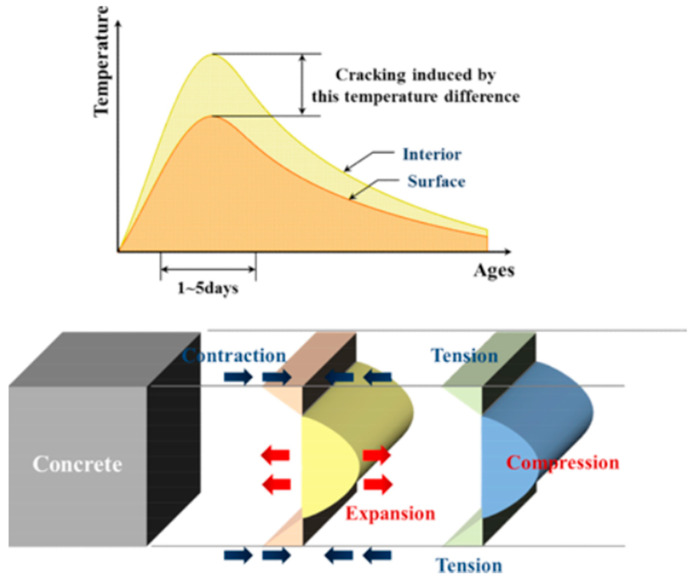
Internal stress induced by temperature gradients [99].

**Figure 6 materials-15-04568-f006:**
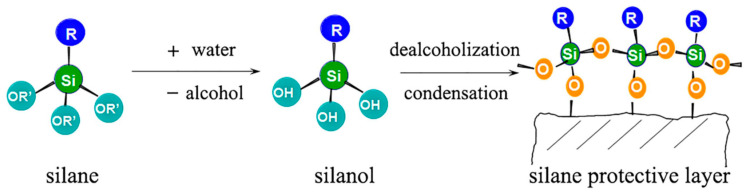
Reaction diagrammatic sketch of silane protective layer [107].

**Figure 7 materials-15-04568-f007:**
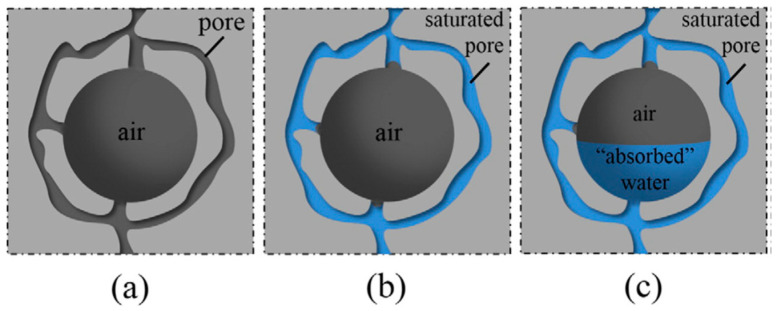
Schematic depicting capillarity and “absorption” of water by air diffusion [109]: (**a**) Before immersion in water; (**b**) Before immersion in water; (**c**) During freezing and thawing.

**Figure 8 materials-15-04568-f008:**
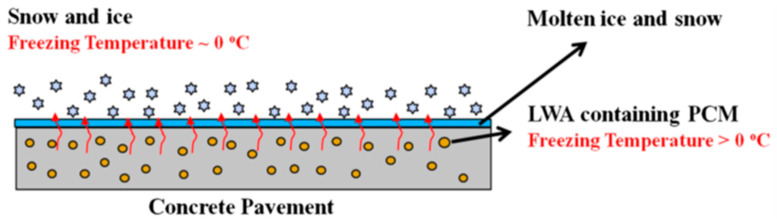
Schematic of using PCM in concrete pavement to melt ice and snow using lightweight aggregate (LWA) [123].

**Figure 9 materials-15-04568-f009:**
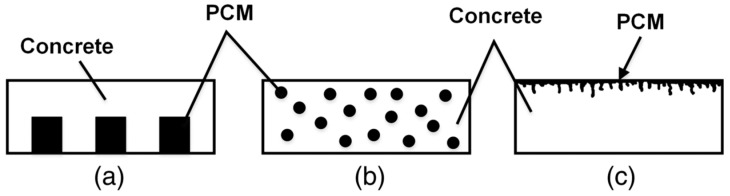
Schematic illustrating three methods of incorporating PCM into concrete: (**a**) using pipes of PCM; (**b**) using particles containing PCM; (**c**) filling concrete surface voids via PCM absorption [123].

**Table 1 materials-15-04568-t001:** Summary on the references related to freeze–thaw damage in concrete pavement.

Pavement Type	Material Type	Location	External Loading	Properties	Model	Reference and Year
Airport pavement	Recycled aggregate concrete	China	Fatigue loading	Compressive strength, flexural strength, relative dynamic modulus	Response surface model	[36] 2018
Airport pavement	Fiber reinforced concrete	China	Wheel impact	Compressive strength, dynamic modulus	-	[37] 2019
Pavement	Ordinary concrete	China	Fatigue loading	Interfacial transition zone	-	[38] 2018
Roller-compacted concrete pavement	Fiber reinforced concrete	Iran	Cyclic loading	Modulus of rupture	-	[39] 2021
Roller-compacted concrete pavement	Ordinary concrete	United States	-	Transverse frequency, mass loss	-	[40] 2005
Airport pavement	Surface-treated concrete	China	-	Dynamic modulus, mass loss	-	[41] 2019
Roller-compacted concrete pavement	Micro-synthetic fiber reinforcement	Iran		Compressive strength, flexural strength, relative dynamic modulus	Response surface model	[42] 2021
Pavement	Jute fiber reinforced concrete	Pakistan	-	Dynamic modulus, mass loss	Empirical equation	[43] 2022
Pavement	Magnesium Phosphate Cement Mortar	China	-	Bonding strength		[44] 2022
Roller-compacted concrete pavement	Concrete containing cement kiln dust	Iran	-	Dynamic modulus	-	[45] 2017
Two-lift concrete pavement	Polypropylene fiber reinforced concrete	China	-	Compressive strength, flexural strength		[46] 2020
Concrete pavement	Concrete with zeolite and metakaolin	Iran	-	Mass loss	-	[47] 2021
Jointed plain concrete pavement	Normal concrete	United States	-	-	Estimation of Concrete Pavement Parameters (ECOPP)	[48] 2018
Concrete pavement	Concrete containing phase change materials	United states	-	Thermal performance	Finite element model	[49] 2019
Concrete pavement	Concrete containing slag aggregate	United states	-	Dynamic modulus	-	[50] 2015

## Data Availability

The data presented in this study are available on request from the corresponding author.

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
