# Peer review of "Impact of Freeze–Thaw Cycles on the Long-Term Performance of Concrete Pavement and Related Improvement Measures: A Review"

_materials, 2022, doi:10.3390/ma15134568_

Round 1

Reviewer 1 Report

Paper entitled “Impact of freeze-thaw cycles on the long-term performance of concrete pavement and related improvement measures: a review” brings an interesting review about freeze-thaw cycles in concrete pavement. Some suggestions for improvements can be seen in the attached file.

Author Response

The detailed response can be found in the attached document.

Reviewer 2 Report

The authors prepared a review paper to study the influence of freeze-thaw cycles on the long-term performance of the concrete pavement. Although the topic is interesting, major revision is necessary to improve the manuscript structure. Hence, some crucial issues need to be addressed before the manuscript is considered for publication. The main comments are mentioned as follows:

- Please provide a schematic figure to explain the different sections considered in the present study.

- Please provide a comprehensive Table to summarize the details of the literature.

- The reviewer recommends to analysis the details of the literature, such as years and regions.

- Page 3, Line 102: Remove “Chapter”.

- Please do not use short paragraphs throughout the manuscript.

- Section 2.5, Page 4, Line 172: Why is the research gap section located here? Please provide a specific section at the end of the manuscript regarding “research gaps and future works”.

- Section 4.3.1 (Page 8, line 348): the reviewer recommends considering a separate section only concentrating on practical methods for delaying F-T damages (like section 5).

- Page 9, Lines 388-392: Please explain more About superabsorbent polymer (SAP), as SAP provides a pore network within the concrete.   For more information regarding air networks using SAP, the authors can use the following reference:

[] Mousavi, S. S., Guizani, L., Bhojaraju, C., & Ouellet-Plamondon, C. (2021). The effect of air-entraining admixture and superabsorbent polymer on bond behaviour of steel rebar in pre-cracked and self-healed concrete. Construction and Building Materials, 281, 122568.

- Regarding using AE, SAP, and PCM, there is a concern about mechanical strength reduction due to the generating of pore network within the matrix. This should be mentioned in the manuscript. The dosage and type of these materials should be checked before using them to delay F-T damages.

Author Response

(The authors gave the same response as above.)

Reviewer 3 Report

Reviewed manuscript titled „Impact of freeze-thaw cycles on the long-term performance of concrete pavement and related improvement measures: a review” is in my opinion a one of the many publications that have considered topic of ordinary concrete frost resistance. The research have not enough additional science knowledge about concrete pavements. Only one paragraph about phase change materials (one page) is form me not enough to title articles about pavement concrete. Authors mainly considered the knowledge from point of view of ordinary concrete – not concrete pavements (only 10 from 72 cited literature have pavement in title.

Prevention of concrete deterioration by superabsorbent polymers has never been widely used in pavements. Phase change materials are quite new idea and this aspect should be extensively investigated from point of view of concrete pavements. This aspect should be extended in article.

Authors should focused only about long-term performance of concrete in real pavements. The laboratory investigation cited in manuscript talk about knowledge known 10 years ago.

Author Response

(The authors gave the same response as above.)

Round 2

Reviewer 1 Report

The authors made significant changes to the paper, making it much better. Virtually, all recommendations have been adjusted accordingly. Therefore, I consider the paper suitable for publication in the journal.

Reviewer 2 Report

The authors appropriately improved the manuscript structure. 

Reviewer 3 Report

The authors improved the article satisfactorily.